# Pore Modification and Phosphorus Doping Effect on Phosphoric Acid-Activated Fe-N-C for Alkaline Oxygen Reduction Reaction

**DOI:** 10.3390/nano11061519

**Published:** 2021-06-08

**Authors:** Jong Gyeong Kim, Sunghoon Han, Chanho Pak

**Affiliations:** Graduate School of Energy Convergence, Institute of Integrated Technology, Gwangju Institute of Science and Technology, Gwangju 61005, Korea; xaso123@gm.gist.ac.kr (J.G.K.); math7103@gm.gist.ac.kr (S.H.)

**Keywords:** oxygen reduction reaction, anion exchange membrane fuel cells, Fe-N-C, phosphoric acid doping, SBA-15

## Abstract

The price and scarcity of platinum has driven up the demand for non-precious metal catalysts such as Fe-N-C. In this study, the effects of phosphoric acid (PA) activation and phosphorus doping were investigated using Fe-N-C catalysts prepared using SBA-15 as a sacrificial template. The physical and structural changes caused by the addition of PA were analyzed by nitrogen adsorption/desorption and X-ray diffraction. Analysis of the electronic states of Fe, N, and P were conducted by X-ray photoelectron spectroscopy. The amount and size of micropores varied depending on the PA content, with changes in pore structure observed using 0.066 g of PA. The electronic states of Fe and N did not change significantly after treatment with PA, and P was mainly found in states bonded to oxygen or carbon. When 0.135 g of PA was introduced per 1 g of silica, a catalytic activity which was increased slightly by 10 mV at −3 mA/cm^2^ was observed. A change in Fe-N-C stability was also observed through the introduction of PA.

## 1. Introduction

The greenhouse effect and climate crisis have driven a transition in energy technology from fossil fuels to renewable energy [1,2,3]. Fuel cells, which are energy conversion devices with high efficiency and low emissions, have attracted immense interest in recent decades [4,5]. However, the sourcing of hydrogen, insufficient hydrogen storage technology, durability problems, and cost issues with noble metal catalysts, such as platinum, have hindered the commercialization of fuel cells [6,7]. The limitations have motivated several studies on lowering the cost of catalysts, which account for 41% of the manufacturing cost of fuel cells, at 500,000 systems per year [8,9]. As the oxygen reduction reaction (ORR) at the cathode is sluggish, it requires a greater catalyst amount compared to the hydrogen oxidation reaction at the anode. Therefore, the major focus has been to reduce the amount of platinum used in the cathode. Development of Pt-based alloys, low-loadings of platinum, modifications of the support, and alternative metal catalysts have been considered [10,11,12,13,14]. For example, alloys of noble metals (Pt, Pd) and transition metals (Cu, Co) have been widely studied [15,16,17,18,19]. Non-noble metal catalysts such as transition metal alloys and metal oxides supported on carbon materials have also been reported [20,21]. In addition, modified carbon with heteroatom doping and porosity control has been shown to have a lower overpotential for ORR than pristine carbon, thus showing great promise for future ORR catalysts [22,23,24,25,26,27]. Alloys and low-loading or modified carbon and non-noble metal approaches have led to improvements in performance, durability, and cost; however, efforts are underway to develop a new type of platinum-free catalyst [28,29,30,31]. The Metal-N-C (M-N-C) catalyst type has recently attracted significant attention. It consists of carbon supports and M-N_x_ active sites advantageous for the ORR [32,33]. As the M-N-C catalysts have higher intrinsic activity than Pt/C catalysts under alkaline conditions, the application of M-N-C to anion exchange membrane fuel cells (AEMFC) is being intensively studied [34,35]. However, due to durability problems with the catalyst and the membrane in single cells, M-N-C catalysts have shown lower performance than Pt/C catalysts [36]. Fortunately, these problems have been solved by the development of new polymer membranes and binders, and a power density of 1.48 W/cm^2^ at 0.6 V was achieved recently [37]. Nevertheless, the intrinsic activity, stability, and electrode structure need to be improved for maximum performance [38,39]. In this regard, doping with different elements, such as phosphorus and sulfur, has been considered, as this can engineer the electronic states of active sites [40,41].

As the M-N_x_ moieties can be formed at defects surrounded by N atoms during pyrolysis, and mesopores are needed for mass transfer, microporous and mesoporous supports such as zeolitic imidazole frameworks have been widely used for both acid and alkaline conditions [42,43,44,45]. Alternatives that immobilize rich M-N_x_ sites include methods that exploit silica-coating or silica templates [46,47]. Joo et al. reported a novel immobilization strategy using silica-coated CNT and found that silica-coating suppressed the growth of nanoparticles, with mainly M-N_x_ moieties remaining after pyrolysis [46]. Notably, impregnation into ordered mesoporous silica resulted in suppressed growth of nanoparticles and agglomerates simultaneously [47]. For these reasons, the application of ordered mesoporous silica for M-N-C catalysts is being considered.

Since it was reported that a dopant that can break the electroneutrality of a carbon matrix can change ORR activity, interest in dopants with lower electronegativity than C such as B and P began to attract attention [48]. In the case of P, with Fe–P bonding, the result of improving ORR activity by effectively controlling the electronic state of the metal is drawing attention [49]. In addition, P–O can cause charge delocalization of the carbon matrix, which leads to the change of the electronegativity of O; therefore, it could be advantageous in adsorbing O_2_ and breaking the double bond of O=O [40,50]. Several studies have been reported using the properties of these P dopants. For example, Guo et al. reported that P, N-doped Co encapsulated carbon nanotubes with Co-P bonding have ORR activity in both acidic and alkaline mediums [51]. Chen et al. developed an active alkaline ORR catalyst by developing N, P co-doped Fe–C composed of a large mesopore and macropore of several tens of nm size using polystyrene as a hard template. However, as previously reported, since the pore distribution can have a major influence on the activity of the ORR catalysis in acidic conditions, further studies on relatively small mesopores (2–10 nm) are required for alkaline conditions [34,52]. Deng et al. reported on the catalysts that used phytic acid as a self-template and P-dopant simultaneously, to synthesize N-P-Fe-tridoped carbon with a hierarchical porous structure to synthesize an ORR catalyst with an excellent half-wave potential of 926 mV at 0.1 M KOH [53]. As described above, studies on P-doped Fe-N-C to improve the activity are actively being conducted, but when a template is used, there is no specific study about the effect of charge delocalization by P and the effect of pore structure by a P precursor.

In this study, a silica-templated Fe-N-C catalyst activated by PA was examined. In a previous study, PA activation of ordered mesoporous carbon resulted in changes in the pore structure [54]. However, a detailed investigation of the structural change and activity of Fe-N-C as an ORR catalyst in alkaline conditions from the addition of PA was not conducted. Therefore, in the present work, the role of PA during M-N-C synthesis and the co-doping effect for ORR were studied. The catalysts were analyzed in detail by X-ray photoelectron spectroscopy (XPS), X-ray diffraction (XRD), electrochemical characterization, scanning electron microscopy (SEM), transmission electron microscopy (TEM) and N_2_ adsorption/desorption techniques. Through these analyses, it was confirmed that the pore structure of the catalyst was modified according to the amount of PA, and a modest change in the ORR activity was observed. Therefore, further heterogeneous element doping studies need to be conducted to elaborate the effect of heteroatom doping in catalyst structures and ORR activity in alkaline media.

## 2. Materials and Methods

### 2.1. Synthesis of Ordered Mesoporous Silica Template, SBA-15

For the grain-type SBA-15, 18 g of Poly(ethylene glycol)-block-poly(propylene glycol)-block-poly(ethylene glycol), denoted as P123 (Average M_n_ = 5800, Sigma-Aldrich, St. Louis, MO, USA) was dispersed in 730 mL of distilled water (DIW), 128.5 mL of HCl (37%, Junsei Chemical, EP, Tokyo, Japan) and 2.16 g of ammonium molybdate tetrahydrate (99%, DUSKAN, EP, Seoul, Korea) mixture. The above solution was heated at 35 °C in a water bath with stirring at 350 rpm for thermal equilibrium for a minimum of 6 h. 42 mL of tetraethyl orthosilicate (95%, SAMCHUN, Seoul, Korea) was added, followed by 5 min of stirring. Then, after leaving it at 35 °C for 24 h, it was moved to an oven at 100 °C for the aging process for an additional 24 h. Thereafter, the solution was taken out, filtered, and washed with DIW and ethanol, and dried overnight at 80 °C. After collecting the white powder that formed, a calcination process was performed at 550 °C for 3 h to completely remove P123 with 3 °C/min of ramping rate.

### 2.2. Synthesis of Fe-N-C Catalysts

First, 0.87 g of 1,10-phenanthroline monohydrate (>99%, DAEJUNG, GR, Seoul, Korea) and phosphoric acid (85%, DAEJUNG, EP, Seoul, Korea) were mixed with 3 g of methanol (DUKSAN, HPLC, Seoul, Korea). The above solution was combined with 1 g of SBA-15 and then dried in a convection oven at 80 °C for 1 h. Then, it was polymerized in an oven at 160 °C for 6 h, and the resulting white powder was collected, which was then mixed with 0.65 g of FeCl_3_ hexahydrate (Sigma-Aldrich, St. Louis, MO, USA) using a mortar and pestle. The yellowish red powder that formed was collected and pyrolyzed to 900 °C at a ramping rate of 2 °C/min under an Ar flow. The temperature was maintained for 3 h, and then cooled to room temperature. The resultant black powder was dispersed in HF (50%, SAMCHUN, EP, Seoul, Korea), diluted with a DIW and ethanol solution (DIW:EtOH = 1:1) to remove silica, stirred for 1 h, and then washed with DIW. It was dried overnight in an oven at 80 °C. To remove unwanted species such as iron oxide and iron nanoparticles, 0.1 g of the powder was dispersed in 100 mL of a 0.5 M sulfuric acid solution, followed by stirring at 80 °C for 8 h. It was filtered, washed with DIW, and pyrolyzed to repair the carbon support structure at 900 °C for 1 h under Ar flow. For the last 20 min, NH_3_ gas was mixed with Ar at the volumetric ratio of 4:1 (Ar:NH_3_). After cooling to room temperature under only Ar flow, it was noted that the product was as Fe-N-C_PA-x, where x is the amount of PA solution added (x: 0.066, 0.100, 0.135) per gram of silica. For comparison, a catalyst without PA was synthesized. The synthesis procedure was the same as above, except for the absence of PA, and it was denoted as Fe-N-C.

### 2.3. Physical Characterization

SEM images were taken using a Hitachi S4700. XRD analysis was performed using the X’Pert PRO Multi-Purpose X-Ray diffractometer (Malvern PANalytical, Malvern, UK). The pore structure of catalysts was investigated by TEM (Tecnai G2 F30 S-Twin, Hillsboro, OR, USA). N_2_ adsorption/desorption experiments were performed on a BELSORP MAX (Microtrac MRB, Osaka, Japan) to measure the specific surface areas and pore structure of silica and carbon. The catalyst samples were pre-treated at 200 °C for 4 h under vacuum before the measurement. The specific surface area was determined by the Brunauer–Emmett–Teller (BET) method. The pore size distribution and pore volume of SBA-15 and Fe-N-Cs were determined by the non-local density functional theory using cylinder (SBA-15) and slit (Fe-N-C) shape pore model, Tikhonov regularization for fitting, and solid definition for pore width. XPS was used to assess the electronic state of the elements at the surface of catalysts by NEXSA (Thermo Fisher Scientific, Waltham, MA, USA).

### 2.4. Electrochemical Characterization

The electrochemical activity of the catalyst was evaluated through a half-cell test using a rotating disk electrode (RDE) technique. A 0.196 cm^2^ glassy carbon electrode was used as the working electrode, and a graphite rod and a Hg/HgO electrode were used as counter and reference electrodes, respectively. The electrolyte was 0.1 M KOH, and the potential relative to Hg/HgO was converted to the RHE (Reversible Hydrogen Electrode) by calibration in hydrogen purged electrolytes. The surface of the glassy carbon was polished with 0.3 μm alumina in order to be mirror-clear. For Fe-N-C, catalyst ink was made by dispersing 7.5 mg of catalyst to 0.1 mL of DIW, 1.23 mL of anhydrous ethyl alcohol (99.9%, DAEJUNG, Seoul, Korea), and 18.8 μL Nafion perfluorinated resin (aqueous dispersion, 10 wt.%, Sigma-Aldrich, St. Louis, MO, USA) This was then sonicated for at least 80 min. 7.8 μL of ink was dropped on the surface of the glassy carbon electrode and 240 μg/cm^2^ of the catalyst was loaded. For comparison 40 wt.% Pt/Vulcan (FC Catalyst, Fuel Cell store, College Station, TX, USA) was used. 3.5 mg of catalyst was dispersed in 0.1 mL of DIW, 1.24 mL of ethyl alcohol anhydrous, and 10.5 μL of Nafion perfluorinated resin. The sonication condition was the same as used for Fe-N-C. Next, 7 μL of ink was coated on the glassy carbon achieving 40 μg_Pt_/cm^2^. Before the ORR activity test, the catalyst was purged with N_2_ and activated by 20 cycles at 100 mV/s in the range of 0.05–1.0 V. The catalyst activity for ORR was measured at a sweep rate of 10 mV/s from 1.2 to 0.1 V while rotating at 1600 rpm with O_2_ purged. An accelerated stress test (AST) was conducted for Fe-N-C_PA-0.135 and 40 wt.% Pt/Vulcan. The potential cycling range was from 0.6 to 1.0 V at 50 mV/s of sweep rate with O_2_ purged for 10,000 cycles and oxygen reduction activity was measured after AST. Solution resistance was compensated by 85% for measuring ORR catalytic activity. This compensation was not applied to the durability test.

## 3. Results and Discussion

Prior to the synthesis of Fe-N-C, the physical properties of SBA-15 used as a sacrificial template were characterized. Based on the SEM image, the silica particles had an ovoid shape with a width and length of approximately 700 nm, and individual particles were separated without agglomeration (Figure 1a). In the XRD pattern, three main peaks were observed near 1°, 1.5°, and 2°, which corresponded to the (100), (110), and (200) planes of the ordered pore arrangement, respectively (Figure 1b) [55]. In the nitrogen isotherm curves, an IUPAC type IV shape was observed, which is typical for mesoporous materials (Figure 1c) [56]. The pore shape was confirmed to be cylindrical, according to the shape of hysteresis [56]. Pores of SBA-15 were mainly composed of mesopores with a size of approximately 12 nm, and micropores or small mesopores with a size close to 2 nm were also found to exist, as displayed in Figure 1d.

After the synthesis of Fe-N-C, according to the method mentioned above, the particle morphologies of the synthesized Fe-N-C catalysts were observed through SEM and TEM (Figure 2). Differences in particle shape and size according to the PA content were negligible in the SEM images. Since a small amount of PA could act as an acid catalyst for polymerization outside the particles, PA was confirmed to activate at the nanometer level [54]. In TEM images, it was confirmed that the pore structure of the carbon derived from SBA-15 consisted of well-arranged straight mesopores. The pore structure did not show any significant difference according to the addition of PA. This suggested that the change in pore structure caused by PA was suppressed by the carbon structure graphitization effect by Fe [57]. In addition, there was no byproduct of large Fe particles, which indicated that silica excluded iron particles and composites by inhibiting the growth of Fe particles [58], as mentioned in the Section 1.

A change in the pore structures of micropores and mesopores was observed through XRD and nitrogen adsorption/desorption measurements. The XRD pattern of Fe-N-C catalysts was measured in two areas. The region less than 10° was for observing a peak due to an ordered pore structure [59]. First, in the case of Fe-N-C without PA, two of three peaks derived from SBA-15 could not be found (Figure 3a). In the nano-replication process, the (110) and (200) peaks, which were smaller than the peak of the (100) plane, were hardly observed. When PA was added, the peak broadened for Fe-N-C_PA-0.066 (Figure 3b). In Fe-N-C_PA-0.100 with a higher PA content, a small (100) peak was observed (Figure 3c). In Fe-N-C_PA-0.135, the peak of the (100) plane was eliminated and a new peak at 2° appeared (Figure 3d). These results indicated that the ordered pore structure of Fe-N-C was modified by the addition of PA. As most of the iron particles or iron compounds were removed by the acid treatment process, only the XRD peak due to carbon was observed. The broad peaks around 23° and 45° were due to the amorphous carbon structure, and the sharp peak around 26° was due to the crystallinity of graphitic carbon [60,61]. Interestingly, the size of the graphitic peak decreased as the PA content increased, and the peak disappeared in the Fe-N-C_PA-0.135 pattern (Figure 3h). It was inferred that PA interfered with the crystallization of carbon while modifying the carbon structure during activation. Thus, the degree of change in the carbon support structure increased as more PA was added.

To analyze the change in pore structure in a quantitative way, the specific surface area, pore volume, and pore size distribution of the Fe-N-C catalysts were determined using nitrogen adsorption/desorption isotherms. All of the obtained N_2_ isotherms had a slit-type pore shape with or without PA (Figure 4a) [56]. The BET specific surface areas were 1070, 1050, 1110, and 1200 m^2^/g for Fe-N-C, Fe-N-C_PA-0.066, Fe-N-C_PA-0.100, and Fe-N-C_PA-0.135, respectively, as listed in Table 1. In the case of Fe-N-C, nanopores with a size of 1 nm to 3 nm developed (Figure 4b).

Interestingly, in the case of the PA-0.066 sample, it was confirmed that the amount of micropore and mesopore decreased compared to Fe-N-C, and then gradually recovered as the phosphoric acid content increased. In addition, in the case of the PA-0.066 sample, pores between 0.8 and 1.0 nm increased, suggesting that pore expansion occurred. However, in the PA-0.100 and PA-0.135 samples, the micropore of 0.6 nm size increased and the mesopore decreased. For the PA-0.066, the PA amount was not sufficient to cover the entire carbon surface; therefore, partial activation of the carbon surface induced the irregular activation of carbon. It was considered that the micropores could not be formed, and rather collapsed, and PA-covered carbon underwent activation with pore expansion. Since PA was not uniformly distributed, therefore, it was expected that added PA excessively existed in some regions, resulting in pore expansion. This can be confirmed in XRD as well. The XRD peak near 1.3° due to the regular pore structure of Fe-NC occurs as the pore structure of PA-0.066 expanded, and the peak broadening occurs in PA-0.100 and PA-0.135. Although there was some scattering, it was confirmed that a peak appeared at a similar location again. Through this, it was confirmed that when PA completely covered the carbon surface, the pore structure changed relatively regularly. Further research is needed on the detailed mechanism and process.

XPS was performed to examine the effect of PA addition on the electronic properties of the Fe-N-C catalyst (Figure 5). Calibration was performed at 284.8 eV based on the C 1s peak. The compositions of the catalysts derived from the XPS survey are presented in Table 2. The N 1s spectra was deconvoluted into pyridinic N (398.2 ± 0.1 eV), Fe-N_x_ (399.8 ± 0.1 eV), pyrrolic N (400.6 ± 0.1 eV), graphitic N (401.2 ± 0.1 eV) and two oxidized N (402.4 ± 0.1 eV and 404.8 ± 0.1 eV) [30,62,63]. Graphitic N was dominant through two heat treatment processes in all catalysts. There was no significant difference among the compositions of N in the catalysts, and only oxidized N with a high oxidation number was observed in some samples, but this was presumed to be due to oxidation in air. P 2p spectra was deconvoluted into P-C (132.1 ± 0.1 eV) and two oxidized P (133.0 ± 0.1 eV and 133.9 ± 0.1 eV) peaks [64,65,66]. In the synthesis method, PA was polymerized with 1,10-phenanthroline; therefore, bare interaction between Fe and P was observed. The change of P species according to the amount of phosphoric acid added was not obviously observed; however, the content of P was in proportion to the amount of phosphoric acid. Fe 2p spectra were deconvoluted with FeO (709.2 eV), Fe-N/Fe_2_O_3_ (711.9 eV), Fe hydroxides (713.8 eV), and satellite peaks (718.9 eV) [67] (Figure 6). Due to the low content of Fe, it was difficult to precisely reveal the electronic state of Fe; this has been reported by previous studies on M-N-C [41,47]. Generally, the peak due to Fe-N or Fe_2_O_3_ was dominant, and the difference between the catalysts was insignificant. The interaction between Fe and P did not appear as in the P 2p spectra. Therefore, it was confirmed that P did not interact with Fe and was simply located in the carbon matrix.

The effect of the analyzed properties of catalysts on the ORR was examined using a half-cell RDE technique. As shown in Figure 7a, all of the Fe-N-C catalysts had higher activity than the Pt/C catalyst in the alkaline medium. The potentials at the kinetic region of catalysis, −3 mA/cm^2^, were 867 mV for Pt/C and 903 mV for Fe-N-C. The potential for Fe-N-C_PA-0.066 was found to be 907 mV, which is slightly higher than that of Fe-N-C. In the case of Fe-N-C_PA-0.100 and Fe-N-C_PA-0.135, the mesopore amount decreased and micropore distribution changed compared to that of Fe-N-C. The potential at 3 mA/cm^2^ was found to be 910 mV and 913 mV, which were 10 mV higher than that of Fe-N-C. The durability of the Fe-N-C, Fe-N-C_PA-0.135 and Pt/C catalysts was tested via AST. After performing 10,000 cycles of AST, potential at −3 mA/cm^2^ of Fe-N-C was negatively shifted by 27 mV, Fe-N-C_PA-0.135 by 28 mV and Pt/C by 41 mV. Through acid treatment and two heat treatment processes, unstable Fe species were removed, and the carbon support was more graphitized, resulting in relatively higher durability than Pt/C. After AST, interestingly, the current of Fe-N-C at mass transport region (@0.6 V) decreased about 15.7%, which was 3.5 and 2.9 times of 4.5% of PA-0.135 and 5.4% of Pt/C, respectively. These results were expected to be due to changes in carbon stability due to P doping. The Fe-N-C_PA-0.135 was found to have high activity in the kinetic region despite the low loading amount compared to the previously reported alkaline ORR catalysts (Figure 7c) [53,68,69,70,71,72,73,74,75,76,77,78]. These results are expected to contribute to solving the water transport problem by electrode thickness [36].

## 4. Conclusions

In summary, a detailed study on the role of PA in the synthesis of Fe-N-C catalysts was conducted. The effect of changes in the pore structure and the change in the electronic state and ORR activity due to P doping were analyzed. Based on the nitrogen adsorption/desorption measurements and XRD analysis, it was concluded that PA alters the pore structure. Interestingly, the activity that was higher by 10 mV was observed in Fe-N-C_PA-0.135, even though the porosity rather decreased compared to Fe-N-C prepared without PA.

As a result of introducing PA into Fe-N-C, changes in the pore structure and P doping were observed. The regular pore structure of Fe-N-C was partially changed by the introduction of PA, and caused a decrease in overall porosity, as investigated from XRD and N_2_ adsorption experiments, respectively. Despite the decrease in porosity, the ORR activity was maintained or showed modest improvement, which is thought to be due to the trade-off of the decrease in porosity and P doping. However, when PA was added, the durability of the catalyst in the mass transport region increased, which showed the possibility that P could stabilize the structure of the Fe-N-C.

## Figures and Tables

**Figure 1 nanomaterials-11-01519-f001:**
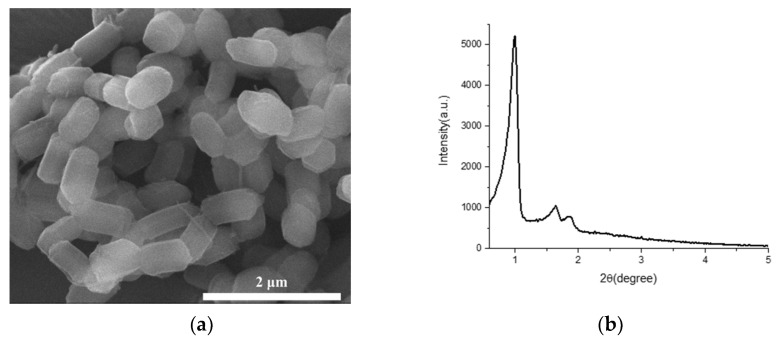
Physical characteristics. (**a**) SEM image; (**b**) X-ray diffraction pattern; (**c**) N_2_ isotherm; (**d**) Pore size distribution of SBA-15.

**Figure 2 nanomaterials-11-01519-f002:**
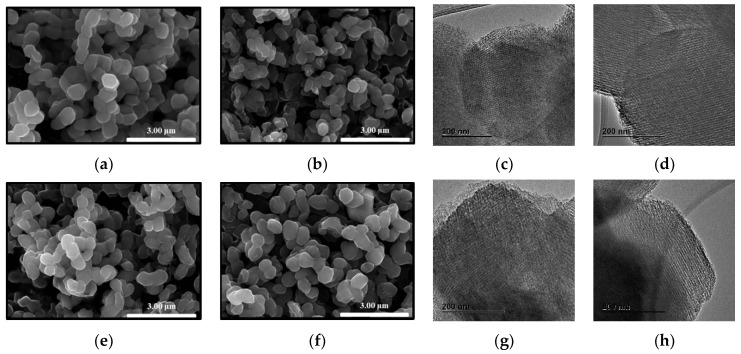
SEM and TEM images of (**a**,**c**) pristine Fe-N-C, (**b**,**d**) Fe-N-C_PA-0.066, (**e**,**g**) Fe-N-C_PA-0.100, and (**f**,**h**) Fe-N_PA-0.135.

**Figure 3 nanomaterials-11-01519-f003:**
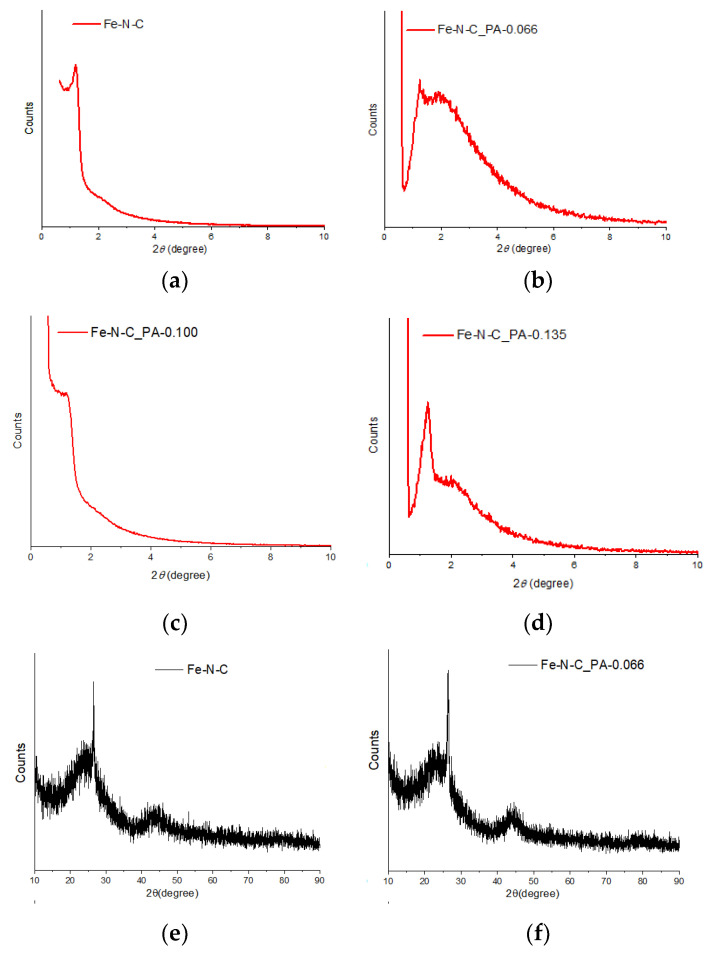
X-ray diffraction patterns of (**a**,**e**) pristine Fe-N-C, (**b**,**f**) Fe-N-C_PA-0.066, (**c**,**g**) Fe-N-C_PA-0.100, and (**d**,**h**) Fe-N_PA-0.135.

**Figure 4 nanomaterials-11-01519-f004:**
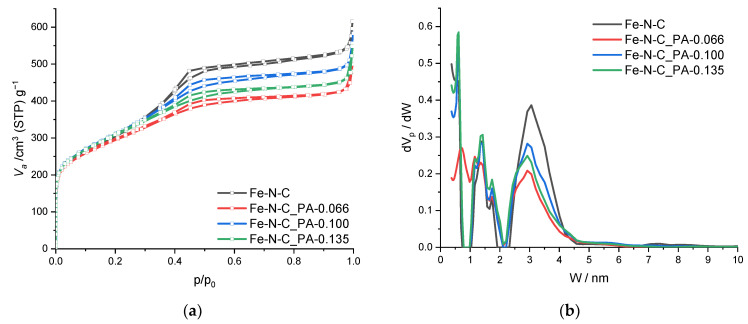
(**a**) N_2_ isotherm and (**b**) pore size distribution of Fe-N-C catalysts. (V_a_: adsorbed volume; dV_p_: differential pore volume; dV_p_/dW: pore area distribution).

**Figure 5 nanomaterials-11-01519-f005:**
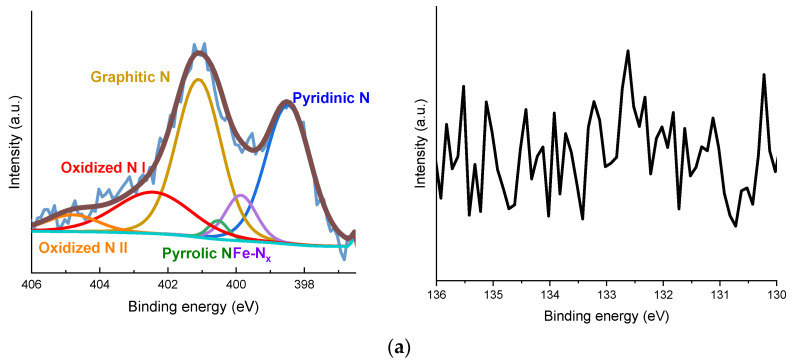
(**a**) N 1 s spectra of Fe-N-C; N 1 s spectra and P 2p spectra of (**b**) Fe-N-C_PA-0.066, (**c**) Fe-N-C_PA-0.100, and (**d**) Fe-N-C_PA-0.135.

**Figure 6 nanomaterials-11-01519-f006:**
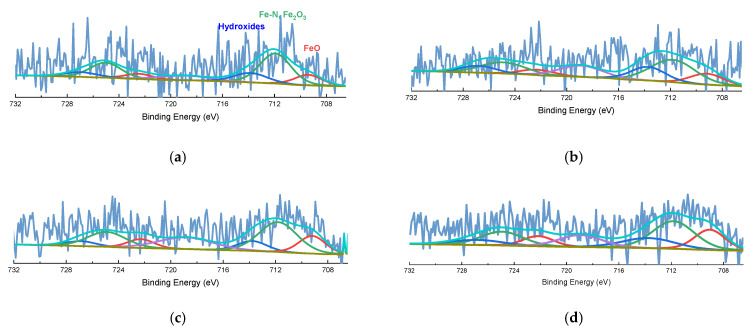
Fe 2p spectra of (**a**) Fe-N-C, (**b**) Fe-N-C_PA-0.066, (**c**) Fe-N-C_PA-0.100, and (**d**) Fe-N-C_PA-0.135.

**Figure 7 nanomaterials-11-01519-f007:**
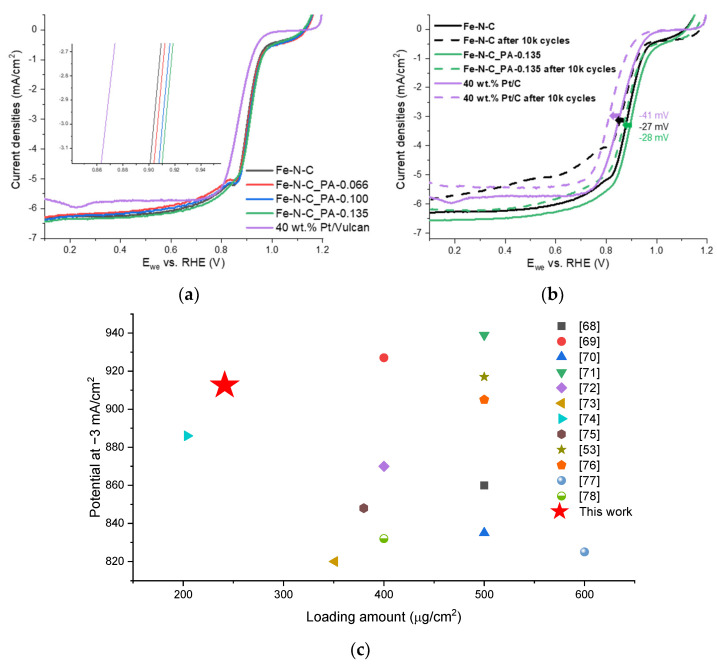
(**a**) Linear sweep voltammograms of Fe-N-C catalysts and Pt/C catalyst; (**b**) durability test result cycling between 0.6 and 1.0 V for 10,000 cycles at a sweep rate of 50 mV/s. The amount of Fe-N-C_PA-0.135 was 480 μg/cm^2^. (**c**) Comparison with reported alkaline ORR Fe-N-C catalysts with potential at −3 mA/cm^2^ along the different catalyst loading amount.

**Table 1 nanomaterials-11-01519-t001:** Specific surface areas and pore volumes of SBA-15 and Fe-N-C catalysts. (S_BET_: BET specific surface area. V_p_: pore volume, V_p micro_: micropore volume, V_p,meso_: mesopore volume).

	S_BET_ (m^2^/g)	V_p_ (cm^3^/g)	V_p,micro_ (cm^3^/g)	V_p,meso_ (cm^3^/g)
SBA-15	680	0.96	0.11	0.85
Fe-N-C	1070	0.83	0.34	0.49
Fe-N-C_PA-0.066	1050	0.60	0.31	0.29
Fe-N-C_PA-0.100	1110	0.68	0.30	0.38
Fe-N-C_PA-0.135	1200	0.69	0.34	0.35

**Table 2 nanomaterials-11-01519-t002:** Compositions of Fe-N-C catalysts from XPS survey.

at%	Fe	P	N	C
Fe-N-C	<0.1	−	3.26	94.44
Fe-N-C_PA-0.066	<0.1	0.18	3.31	93.32
Fe-N-C_PA-0.100	<0.1	0.29	3.15	93.27
Fe-N-C_PA-0.135	<0.1	0.56	3.20	92.79

## Data Availability

Not Available.

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
