# Peer review of "Pore Modification and Phosphorus Doping Effect on Phosphoric Acid-Activated Fe-N-C for Alkaline Oxygen Reduction Reaction"

_nanomaterials, 2021, doi:10.3390/nano11061519_

Round 1

Reviewer 1 Report

The paper presents the effects of phosphoric acid activation and phosphorus doping using Fe-N-C catalysts and SBA-15 on alkaline oxygen reduction reaction. The methods' presentation and scientific results are satisfactory for publication in the Nanomaterials journal. The minor and major drawbacks to be addressed can be specified as follows:
1.    Page 1, line 22, Keywords. (i) AEMFC ---> anion exchange membrane fuel cells (ii) phosphoric acid ---> phosphoric doping (iii) please add ”SBA-15”.
2.    Page 2, lines 57 and 60. M-nx - should x be written as subscript? Or not? Please standardize.
3.    Page 3, line 123. pore volume ---> pore structure.
4.    Page 4, Fig. 1. Is a legend needed? How did the authors assess the correctness of the synthesis of SBA-15? Comparison with other published data? For example, https://www.acsmaterial.com/sba-15-20g.html. Other references? What pore shape was used to determine the pore size distribution? In my opinion PSD plot is slightly strange. Please discuss this problem in the reviewed paper.
5.    Page 5, Fig. 2(h). Why the resolution in Fig. 2(h) is different than on the other panels (c, d, and g)?
6.    Page 7, Fig. 4(a). Is the y-axis scale is this figure arbitrary? The data collected in Table 1 (Vp) indicate this. It better not be. At present, it makes no sense to mention a unit. Currently, it isn't straightforward to compare the sorption capacity of the tested materials. In conclusion, please do not use arbitrary units!!!
7.    Pages 8-10, Figs. 5 and 6. The theoretical lines describing the experimental data are too thick compared to those relating to the experimental measurements.
8.    Page 9, Tab. 2. (i) Did the authors determine phosphorus content using elemental analysis? XPS measurements are not entirely reliable!!! (ii) Why is the phosphorus content so low for sample Fe-N-C_PA-0.100?
9.    Page 11, Fig. 7(b). (i) Did the authors measure (durability test) the remaining samples (i.e. Fe-N-C_PA-0.066 and Fe-N-C_PA-0.100)? In my opinion, the results presented in Fig. 7 are essential for work. It is necessary to show the results for the remaining samples. It is, in fact, the biggest novelty in this article. Did the authors achieve their goal? Because as one can see, phosphorus content on alkaline oxygen reduction reaction is minimal.

Author Response

Reviewer 1

Before responding to valuable advice from reviewers, we would like to inform you that there are several changes to the manuscript.

  1. First, it is about the N2 adsorption experiment. In the N2 adsorption experiment, we first reported that Fe-N-C_PA-0.066 had a very large amount of micropores below 1 nm, but this result was confirmed to be due to a practically wrong measurement method. To revise this, we re-measured in an appropriate mode for observing low-pressure region precisely for all samples, and revised Figure 4, Table 1, and related contents. In the isotherm curves, the arbitrary unit of the y-axis has been modified to a specific unit.
  2. In the case of XPS, first of all, the clarity of the XPS was improved by identifying the raw data of the XPS and the thickness of the theoretical line. In addition, to solve the problem for deconvolution due to high noise in the P 2p spectra, we tried to improve this through a small step size and a large number of scan numbers. In the XPS survey, we made efforts to measure P content more accurately. These results are reflected in Figure 5 and Table 2, and related contents have also been changed.
  3. The catalytic activity of ORR for all Fe-N-Cs was re-measured, and the durability test for Fe-N-C was additionally carried out, which was reflected in Figure 7 and related paragraphs were revised.
  4. Keywords, subscripts, strange sentences were revised, and according to the reviewer's advice, the activity was compared with the non-precious metal catalysts for alkaline ORR previously reported in Figure 7, and the description of the difference from other studies was reinforced in the Introduction.

We would be grateful if you would review the manuscript referring to the above four points.

------------------------------------------------------------------------------------------------------------------------

The paper presents the effects of phosphoric acid activation and phosphorus doping using Fe-N-C catalysts and SBA-15 on alkaline oxygen reduction reaction. The methods' presentation and scientific results are satisfactory for publication in the Nanomaterials journal. The minor and major drawbacks to be addressed can be specified as follows:
1.    Page 1, line 22, Keywords. (i) AEMFC à anion exchange membrane fuel cells (ii) phosphoric acid à phosphoric doping (iii) please add ”SBA-15”.

Response: Thanks for your thoughtful suggestions. We took your comments carefully and have revised the keywords of our manuscript.

Page 1, Lines 20-21;

Keywords: oxygen reduction reaction; anion exchange membrane fuel cells; Fe-N-C; phosphoric acid doping; SBA-15

  1.    Page 2, lines 57 and 60. M-Nx - should x be written as subscript? Or not? Please standardize.

Response: We appreciate your comment. We intended x to be subscript as M-Nx. We corrected this issue for the entire manuscript.

Page 2, Lines 1, 12, 15, 18; Page 8, Line 5; Page 11, Line 19

M-Nx and Fe-Nx were converted into M-Nx and Fe-Nx.

  1.    Page 3, line 123. pore volume à pore structure.

Response: Thank you for the recommendation for the proper word. We corrected with it.

Page 3, Line 46;

the specific surface areas and pore structure of silica and carbon

  1.    Page 4, Fig. 1. Is a legend needed? How did the authors assess the correctness of the synthesis of SBA-15? Comparison with other published data? For example, https://www.acsmaterial.com/sba-15-20g.html. Other references? What pore shape was used to determine the pore size distribution? In my opinion PSD plot is slightly strange. Please discuss this problem in the reviewed paper.

Response: We are grateful for your helpful comments. In Fig. 1, we also thought that the legend was unnecessary, therefore, we deleted them. Synthesized SBA-15 was compared to the previous works on SBA-15 with appropriate references [R1]. For the low angle region of XRD, as we wrote in the manuscript, the ordered pore structure of SBA-15 was confirmed from the main peak and two smaller peaks, which are assigned to (100), (110), (200) planes of the ordered pore arrangement which is the characteristics of SBA-15. And in the case of SBA-15, the pore structure and particle shape may change as the synthesis parameters change. For pore size distribution, we used cylinder shape pore model and with Tikhonov regularization for fitting with a solid definition of pore width for SBA-15.

Page 3, Lines 48-51;

The pore size distribution and pore volume of SBA-15 and Fe-N-Cs were determined by the non-local density functional theory using cylinder (SBA-15) and slit (Fe-N-C) shape pore model, Tikhonov regularization for fitting, and solid definition for pore width.

  1.    Page 5, Fig. 2(h). Why the resolution in Fig. 2(h) is different than on the other panels (c, d, and g)?

Response: Thanks for your important question. We tried to show TEM images that could effectively show the ordered mesopores. However, as you mentioned, comparing at the same resolution can provide more readability, therefore, we changed these to images at the same resolution.

  1.    Page 7, Fig. 4(a). Is the y-axis scale is this figure arbitrary? The data collected in Table 1 (Vp) indicate this. It better not be. At present, it makes no sense to mention a unit. Currently, it isn't straightforward to compare the sorption capacity of the tested materials. In conclusion, please do not use arbitrary units!!!

Response: We are grateful for your thoughtful advice. It was our mistake for deleting the y-axis number during configuring figures. We corrected the entire Fig.4 without an arbitrary unit. Thank you for your sensitive observation again.

Page 7, Line 22;

Figure 4 was corrected with a number using a specific unit.

  1.    Pages 8-10, Figs. 5 and 6. The theoretical lines describing the experimental data are too thick compared to those relating to the experimental measurements.

Response: Thank you for your thoughtful comment. We synchronized the thickness of raw data and deconvolution lines for better recognition for Fig. 5 and 6.

  1.    Page 9, Tab. 2. (i) Did the authors determine phosphorus content using elemental analysis? XPS measurements are not entirely reliable!!! (ii) Why is the phosphorus content so low for sample Fe-N-C_PA-0.100?

Response: We are really grateful for your critical question. We agree with the opinion that the trend towards added PA amount and P content is not adequate. Therefore, we tried to obtain accurate data by implementing the small step size and sufficiently large scan number when measuring the XPS survey and resulted in different contents of P from previous values. Unfortunately, the determination of the P content through elemental analysis was not conducted. Instead, we tried to determine P content through the inductively coupled plasma spectroscopy, but P was not detected, thus, we are in the process of optimizing the sample pretreatment method such as using strong oxidizing agents. We are very grateful for your suggestion and will reflect it in future works.

  1.    Page 11, Fig. 7(b). (i) Did the authors measure (durability test) the remaining samples (i.e. Fe-N-C_PA-0.066 and Fe-N-C_PA-0.100)? In my opinion, the results presented in Fig. 7 are essential for work. It is necessary to show the results for the remaining samples. It is, in fact, the biggest novelty in this article. Did the authors achieve their goal? Because as one can see, phosphorus content on alkaline oxygen reduction reaction is minimal.

Response: We appreciate your kind opinion. We did not try the durability test for PA-0.066 and PA-0.100. However, after receiving your comment, we conducted a durability test on Fe-N-C instead, since we thought that the effect of P on the stability of catalyst could be certainly observed by comparing Fe-N-C and Fe-N-C_PA-0.135. As a result, it showed an interesting result. There was not a significant deviation in the kinetic region, however, Fe-N-C showed severe degradation in limiting current region. The causes of this are not clear, however, we infer that P stabilized carbon structure or anti-poisoning effect of P [R2]. Therefore, more studies would be needed to reveal it.

Page 10, Lines 3-22;

The effect of the analyzed physical and electronic properties on the ORR was examined using a half-cell RDE technique. As shown in Figure 7a, all the Fe-N-C catalysts had higher activity than the Pt/C catalyst in the alkaline medium. The potentials at the kinetic region of catalysis, -3 mA/cm2, were 867 mV for Pt/C and 903 mV for Fe-N-C. The potential for Fe-N-C_PA-0.066 was found to be 907 mV, which is slightly higher than that of Fe-N-C. Even though the decrease in micropores and mesopores, the effect of P for surface wettability and electronic state was prominent. In the case of Fe-N-C_PA-0.100 and Fe-N-C_PA-0.135, the mesopore amount decreased and micropore distribution changed compared to that of Fe-N-C. The potential at 3 mA/cm2 was found to be 910 mV and 913 mV, which were higher than that of Fe-N-C. The obtained results also showed that the electronic state of the active site can be shifted only by doping P in the carbon plane without bonding with Fe. The durability of the Fe-N-C, Fe-N-C_PA-0.135, and Pt/C catalysts was tested via AST. After performing 10,000 cycles of AST, the potential at -3 mA/cm2 of Fe-N-C was negatively shifted by 27 mV, Fe-N-C_PA-0.135 by 28 mV, and Pt/C by 41 mV. Through acid treatment and two heat treatment processes, unstable Fe species were removed, and the carbon support was more graphitized, resulting in relatively high durability than Pt/C. After AST, interestingly, the current of Fe-N-C at the mass transport region (@0.6 V) decreased about 15.7%, which was 3.5 and 2.9 times of 4.5% of PA-0.135 and 5.4% of Pt/C, respectively. These results were expected to be due to changes in carbon structure stability due to P doping.

References

[R1] Joo, S. H.; Pak, C.; You, D. J.; Lee, S. A.; Lee, H. I.; Kim, J. M.; Chang, H.; Seung, D. Ordered Mesoporous Carbons (OMC) as Supports of Electrocatalysts for Direct Methanol Fuel Cells (DMFC): Effect of Carbon Precursors of OMC on DMFC Performances. Electrochim. Acta 2006, 52 (4), 1618–1626. https://doi.org/10.1016/j.electacta.2006.03.092

[R2] Najam, T.; Shah, S.S.A.; Ding, W.; Wei, Z. Role of P-doping in Antipoisoning: Efficient MOF-Derived 3D Hierarchical Architectures for the Oxygen Reduction Reaction. J. Phys. Chem. C 2019, 123 (27), 16796-16803. https://doi.org/10.1021/acs.jpcc.9b03730

Reviewer 2 Report

The paper, “Pore Modification and Phosphorus Doping Effect on Phosphoric Acid-activated Fe-N-C for Alkaline Oxygen Reduction Reaction”,  reports the catalytic activity improvement in Fe-N-C material after treating with Phosphoric acid.

The authors build on their previous work on ordered mesoporous carbons which were prepared by using sacrificial silica templates. In this work, they have doped such a material with Fe, N and further with Phosphorus and studied it as a replacement for Pt ORR catalysts. Since there are many papers on this subject in literature, the authors must stress the significance of their work better. This could be done either in introduction or by a comparative table (comparing literature results) in the results sections.   For example, there is a 2015 paper on similar topic, “ Promotional effect of phosphorus doping on the activity of the Fe-N/C catalyst for the oxygen reduction reaction”, https://doi.org/10.1016/j.electacta.2014.12.163. Here, the authors attribute in improvement of ORR to phosphorous doping, where they have achieved 2-3% doping. How does the present work improve ORR activity over this article particularly when P doping is very minute at 0.25 at.%?

This half-cell measurements show that the catalytic activity of Fe-N-C_PA-0.100 and 0.135 are slightly better than pristine Fe-N-C sample as well as the lightly doped samples. However, I believe these are modest improvements and importantly the conclusions do not follow the results as of yet, only an inference regarding shifting of electronic states is made.

All doped samples have lower mesopore volume than pristine sample while the micropore volume are comparable. The high micropore volume and surface area of the 0.066 doped sample must be carried out again, including BET, RDE measurements. This sample’s values are quite different from others – it could be a one sample that behaved differently. Hence, differences in pore volume is not sufficient to explain improvement in catalytic activity.

The phosphorous doping concentration is also not consistent, since the 0.100 doped sample has only 0.08 at.% doping compared to others, which are at 0.25 to 0.29%. However the catalytic activity shown by RDE measurements are similar for 0.1 and 0.135 sample. In general, the XPS data is very noisy too. I would suggest redoing the XPS measurements if possible.

What is meant by graphitization of Fe? Do the authors mean crystallization? Or graphitization of the structure?

Overall, I would kindly suggest the authors to repeat the measurements of at least 0.066 and 0.1 doped samples, to confirm their observations and further, provide a better reasoning for the improvement in catalytic activity.

Author Response

Reviewer 2

Before responding to valuable advice from reviewers, we would like to inform you that there are several changes to the manuscript.

  1. First, it is about the N2 adsorption experiment. In the N2 adsorption experiment, we first reported that Fe-N-C_PA-0.066 had a very large amount of micropores below 1 nm, but this result was confirmed to be due to a practically wrong measurement method. To revise this, we re-measured in an appropriate mode for observing low-pressure region precisely for all samples, and revised Figure 4, Table 1, and related contents. In the isotherm curves, the arbitrary unit of the y-axis has been modified to a specific unit.
  2. In the case of XPS, first of all, the clarity of the XPS was improved by identifying the raw data of the XPS and the thickness of the theoretical line. In addition, to solve the problem for deconvolution due to high noise in the P 2p spectra, we tried to improve this through a small step size and a large number of scan numbers. In the XPS survey, we made efforts to measure P content more accurately. These results were reflected in Figure 5 and Table 2, and related contents have also been changed.
  3. The catalytic activity of ORR for all Fe-N-Cs was re-measured, and the durability test for Fe-N-C was additionally carried out, which was reflected in Figure 7 and related paragraphs were revised.
  4. Keywords, subscripts, strange sentences were revised, and according to the reviewer's advice, the activity was compared with the non-precious metal catalysts for alkaline ORR previously reported in Figure 7, and the description of the difference from other studies was reinforced in the Introduction.

We would be grateful if you would review the manuscript referring to the above four points.

------------------------------------------------------------------------------------------------------------------------

The paper, “Pore Modification and Phosphorus Doping Effect on Phosphoric Acid-activated Fe-N-C for Alkaline Oxygen Reduction Reaction”, reports the catalytic activity improvement in Fe-N-C material after treating with Phosphoric acid.

The authors build on their previous work on ordered mesoporous carbons which were prepared by using sacrificial silica templates. In this work, they have doped such a material with Fe, N and further with Phosphorus and studied it as a replacement for Pt ORR catalysts.

  1. Since there are many papers on this subject in literature, the authors must stress the significance of their work better. This could be done either in introduction or by a comparative table (comparing literature results) in the results sections.

Response: Thank you for the helpful comment. We take your opinion very seriously. We have made changes in the direction of emphasizing the necessity and importance of our work by revising Introduction section and adding a comparative table in the results section (Figure 7c).

Page 2, Lines 22-42;

Since it was reported that a dopant that can break the electroneutrality of carbon matrix can change ORR activity, interest in dopants with lower electronegativity than C such as B and P began to attract attention [48]. In the case of P, with Fe-P bonding, the result of improving ORR activity by effectively controlling the electronic state of the metal is drawing attention [49]. In addition, P-O can cause charge delocalization of the carbon matrix, which leads to the change of the electronegativity of O, therefore, it could be advantageous in adsorbing O2 and breaking the double bond of O=O [40,50]. Several studies have been reported using the properties of these P dopants. For example, Guo et al. reported that P, N-doped Co encapsulated carbon nanotubes with Co-P bonding have ORR activity in both acidic and alkaline medium [51]. Chen et al. developed an active alkaline ORR catalyst by developing N, P co-doped Fe-C composed of a large mesopore and macropore of several tens of nm size using polystyrene as a hard template. However, as previously reported, since the pore distribution can have a major influence on the activity of the ORR catalysis in acidic condition, further studies on relatively small mesopores (2-10 nm) are required for alkaline condition [34, 52]. Deng et al. reported the catalysts that using phytic acid as a self-template and P-dopant simultaneously to synthesize N-P-Fe-tridoped carbon with a hierarchical porous structure to synthesize an ORR catalyst with an excellent half-wave potential of 926 mV at 0.1 M KOH [53]. As described above, studies on P-doped Fe-N-C to improve the activity are actively being conducted, but when a template is used, there is no specific study about the effect of charge delocalization by P and the effect of pore structure by P precursor.

Page 10, Lines 22-25;

The Fe-N-C_PA-0.135 was found to have high activity in the kinetic region despite the low loading amount compared to the previously reported alkaline ORR catalysts (Figure 7c). These results are expected to contribute to solving the problem of electrode thickness, one of the chronic problems of AEMFC.

  1. For example, there is a 2015 paper on similar topic, “ Promotional effect of phosphorus doping on the activity of the Fe-N/C catalyst for the oxygen reduction reaction”, https://doi.org/10.1016/j.electacta.2014.12.163. Here, the authors attribute in improvement of ORR to phosphorous doping, where they have achieved 2-3% doping. How does the present work improve ORR activity over this article particularly when P doping is very minute at 0.25 at.%?

Response: Thank you for your important question. The content of phosphorus in the carbon matrix can vary depending on synthesis parameters, such as the pyrolysis temperature, time, and the amount of P precursor added. In the case of our work, catalysts were confirmed to have a very low concentration of P because it was pyrolyzed at 900 °C for 3 hours and underwent additional heat treatment for 1 h after acid leaching. And since the content of Fe (<0.3 at%), as well as P is very low, we consider that there is a possibility that P can change the electronic state of Fe when P is located near Fe even without direct bonds. And for the issue that the P content and the precursor content did not match, the XPS measuring condition was optimized and measurement was performed again, it was confirmed that the amount of P precursor and the P content showed the proper tendency. When more than 0.135 g of P precursor was added, several particles suspected of iron phosphate were formed and the activity decreased. Therefore, we considered 0.135 g as the optimum.

Page 9, Line 8;

  1. This half-cell measurements show that the catalytic activity of Fe-N-C_PA-0.100 and 0.135 are slightly better than pristine Fe-N-C sample as well as the lightly doped samples. However, I believe these are modest improvements and importantly the conclusions do not follow the results as of yet, only an inference regarding shifting of electronic states is made.

Response: We are grateful for your valuable comment. As you mentioned, the analysis of the electronic state of Fe has not been performed yet. Considering the Fe particle size, Fe content, and noise in XPS, more accurate techniques such as X-ray absorption spectroscopy and Mössbauer spectroscopy are needed. However, since we are in a difficult situation to measure these, an estimation based on electrochemical data was inevitably required. However, even though the micropores and mesopores decreased compared to the sample without phosphoric acid, the ORR activity increased. Therefore, it seemed reasonable to assume that a change occurred in the active sites. This is presumed to be the result of the complexes of the pore change caused by phosphoric acid and the doping effect of phosphorus. We gratefully accept your comments and will work on further researches to reveal the electronic state of Fe.

  1. All doped samples have lower mesopore volume than pristine sample while the micropore volume are comparable. The high micropore volume and surface area of the 0.066 doped sample must be carried out again, including BET, RDE measurements. This sample’s values are quite different from others – it could be a one sample that behaved differently. Hence, differences in pore volume is not sufficient to explain improvement in catalytic activity.

Response: We really appreciate your suggestions. This was entirely due to a mistake in our measurement process, and it was confirmed that there was an error in the measurement in the low-pressure region in the N2 adsorption experiment. We performed N2 adsorption again through the correct measurement method and revised Fig. 4. As for the pore distribution, there was a point that we erroneously judged, and related matters have been corrected. In the case of RDE, the ORR activity was measured by newly synthesized under the same conditions, and an increase in activity of 3-4 mV was observed in PA-0.066 and PA-0.135. Therefore, ORR activity was proportional to P content. It was different from our previous data (P-0.066<Fe-N-C<PA-0.100<PA-0.135, before). This might be due to errors such as the condition of the working electrode. To minimize this error, we tried more measurements using a clean working electrode with polishing thoroughly for each catalyst for reproducible results.

Page 10, Lines 3-22;

The effect of the analyzed physical and electronic properties on the ORR was examined using a half-cell RDE technique. As shown in Figure 7a, all the Fe-N-C catalysts had higher activity than the Pt/C catalyst in the alkaline medium. The potentials at the kinetic region of catalysis, -3 mA/cm2, were 867 mV for Pt/C and 903 mV for Fe-N-C. The potential for Fe-N-C_PA-0.066 was found to be 907 mV, which is slightly higher than that of Fe-N-C. Even though the decrease in micropores and mesopores, effect of P for surface wettability and electronic state was prominent. In the case of Fe-N-C_PA-0.100 and Fe-N-C_PA-0.135, the mesopore amount decreased and micropore distribution changed compared to that of Fe-N-C. The potential at 3 mA/cm2 was found to be 910 mV and 913 mV, which were higher than that of Fe-N-C. The obtained results also showed that the electronic state of the active site can be shifted only by doping P in the carbon plane without bonding with Fe. The durability of the Fe-N-C, Fe-N-C_PA-0.135 and Pt/C catalysts was tested via AST. After performing 10,000 cycles of AST, potential at -3 mA/cm2 of Fe-N-C was negatively shifted by 27 mV, Fe-N-C_PA-0.135 by 28 mV and Pt/C by 41 mV. Through acid treatment and two heat treatment processes, unstable Fe species were removed, and the carbon support was more graphitized, resulting in relatively high durability than Pt/C. After AST, interestingly, the current of Fe-N-C at mass transport region (@0.6 V) decreased about 15.7%, which was 3.5 and 2.9 times of 4.5% of PA-0.135 and 5.4% of Pt/C, respectively. These results were expected to be due to changes in carbon structure stability due to P doping.

  1. The phosphorous doping concentration is also not consistent, since the 0.100 doped sample has only 0.08 at.% doping compared to others, which are at 0.25 to 0.29%. However the catalytic activity shown by RDE measurements are similar for 0.1 and 0.135 sample. In general, the XPS data is very noisy too. I would suggest redoing the XPS measurements if possible.

Response: Thank you for your helpful comment. Receiving your comments seriously, we re-measured RDE and XPS. It was confirmed that FeNC_PA-0.100 and FeNC_PA-0.135 did not show a significant difference in terms of activity. This is thought to be caused by the insufficient number of Fe in the carbon support. Since there are low contents of active sites, it is considered that P already contains a sufficient amount of P to change the electronic state of Fe in FeNC_PA-0.100. We are also deeply considering ways to increase the number of Fe active sites of the catalysts to overcome these limitations.

Page 9, Line 8;

  1. What is meant by graphitization of Fe? Do the authors mean crystallization? Or graphitization of the structure?

Response: We are very sorry for the misunderstanding word use. We intended that Fe graphitized the carbon structure. Fe induces the graphitization of the carbon structure in the carbonization process, therefore, materials such as FeCl2 are often used as graphitizing agents. We have tried to improve readability by correcting mistakes like this.

Page 5, Line 12-13;

This suggested that the change in pore structure caused by PA was suppressed by the carbon structure graphitization effect by Fe.

  1. Overall, I would kindly suggest the authors to repeat the measurements of at least 0.066 and 0.1 doped samples, to confirm their observations and further, provide a better reasoning for the improvement in catalytic activity.

Response: Thank you very much for your kind suggestion. As responded to in comment #5, we re-tested XPS and RDE and corrected the figures and tables which reflect these results. In addition, we have modified and added the discussion accordingly. We will be grateful if you consider this for reviewing our revised manuscript.

Page 8, Line 23 - Page 10, Line 25;

Reviewer 3 Report

Similar works have been performed in the last 6 years or so. The following refs are not cited and they show similar studies: Y. Chen et al., J. Nanopart. Res. 2021, 23, 68; J. Li et al., Sci. China Mater. 2020, 63, 965; Y. Hu et al., Electr. Acta 2015, 155, 335. In addition, the xps data show spectra with an appreciable noise, thus deconvolution data do not seem accurate. I recommend not acceptance of this manuscript at this form.

Author Response

Reviewer 3

Before responding to valuable advice from reviewers, we would like to inform you that there are several changes to the manuscript.

  1. First, it is about the N2 adsorption experiment. In the N2 adsorption experiment, we first reported that Fe-N-C_PA-0.066 had a very large amount of micropores below 1 nm, but this result was confirmed to be due to a practically wrong measurement method. To revise this, we re-measured in an appropriate mode for observing low-pressure region precisely for all samples, and revised Figure 4, Table 1, and related contents. In the isotherm curves, the arbitrary unit of the y-axis has been modified to a specific unit.
  2. In the case of XPS, first of all, the clarity of the XPS was improved by identifying the raw data of the XPS and the thickness of the theoretical line. In addition, to solve the problem for deconvolution due to high noise in the P 2p spectra, we tried to improve this through a small step size and a large number of scan numbers. In the XPS survey, we made efforts to measure P content more accurately. These results were reflected in Figure 5 and Table 2, and related contents have also been changed.
  3. The catalytic activity of ORR for all Fe-N-Cs was re-measured, and the durability test for Fe-N-C was additionally carried out, which was reflected in Figure 7 and related paragraphs were revised.
  4. Keywords, subscripts, strange sentences were revised, and according to the reviewer's advice, the activity was compared with the non-precious metal catalysts for alkaline ORR previously reported in Figure 7, and the description of the difference from other studies was reinforced in the Introduction.

We would be grateful if you would review the manuscript referring to the above four points.

------------------------------------------------------------------------------------------------------------------------

  1. Similar works have been performed in the last 6 years or so. The following refs are not cited and they show similar studies: Y. Chen et al., J. Nanopart. Res. 2021, 23, 68; J. Li et al., Sci. China Mater. 2020, 63, 965; Y. Hu et al., Electr. Acta 2015, 155, 335.

Response: We really appreciate your valuable opinion. We performed the XPS experiment on the catalysts again and corrected the data. In the aspect of the novelty of this work, however, I would like to mention a few things that there are differences between the papers you suggested and our work. First, [Y. Chen et al., J. Nanopart. Res. 2021, 23, 68] could be considered similar to our study. Since a hierarchical pore structure formed from polystyrene particles and Fe-N-C doped with P for oxygen reduction reaction was reported. However, there is a difference between the roles of these large mesopores (20-50 nm) even with macropores (>50 nm) and small mesopores (2-10 nm) under acidic condition [R1]. We think that the role of small mesopores for ORR in alkaline medium is required to be studied. For [J. Li et al., Sci. China Mater. 2020, 63, 965] and [Y. Hu et al., Electr. Acta 2015, 155, 335], the target reaction is ORR, but there is a significant difference in pH conditions. In the case of these papers, ORR in acidic condition was measured. Since the ORR in acidic condition and alkaline condition differs in the aspects of the mechanism and product, it is difficult to compare the ORR in acidic condition and the ORR in alkaline condition with the same standards. Of course that there have been many studies on P-doped Fe-N-C catalysts. However, further research is still needed to study the effect of pore structure, electronic state change, and catalytic activity of P-doped Fe-N-C synthesized using ordered mesoporous silica as a hard template. Modifications have been made to the Introduction part to reinforce the explanation of the importance of our research. This manuscript could be further developed by accepting your comments seriously and we deeply appreciate this.

Page 2, Lines 22-42;

Since it was reported that a dopant that can break the electroneutrality of carbon matrix can change ORR activity, interest in dopants with lower electronegativity than C such as B and P began to attract attention [48]. In the case of P, with Fe-P bonding, the result of improving ORR activity by effectively controlling the electronic state of the metal is drawing attention [49]. In addition, P-O can cause charge delocalization of the carbon matrix, which lead to the change of the electronegativity of O, therefore, it could be advantageous in adsorbing O2 and breaking the double bond of O=O [40,50]. Several studies have been reported using the properties of these P dopants. For example, Guo et al. reported that P, N-doped Co encapsulated carbon nanotubes with Co-P bonding have ORR activity in both acidic and alkaline medium [51]. Chen et al. developed an active alkaline ORR catalyst by developing N, P co-doped Fe-C composed of a large mesopore and macropore of several tens of nm size using polystyrene as a hard template. However, as previously reported, since the pore distribution can have a major influence on the activity of the ORR catalysis in acidic condition, further studies on relatively small mesopores (2-10 nm) are required for alkaline condition [34, 52]. Deng et al. reported the catalysts that using phytic acid as a self-template and P-dopant simultaneously to synthesize N-P-Fe-tridoped carbon with a hierarchical porous structure to synthesize an ORR catalyst with an excellent half-wave potential of 926 mV at 0.1 M KOH [53]. As described above, studies on P-doped Fe-N-C to improve the activity are actively being conducted, but when a template is used, there is no specific study about the effect of charge delocalization by P and the effect of pore structure by P precursor.

  1. In addition, the xps data show spectra with an appreciable noise, thus deconvolution data do not seem accurate. I recommend not acceptance of this manuscript at this form.

Response: We are grateful for your valuable comment. As mentioned above, we were advised to re-measure XPS spectra due to noises. Therefore, P 2p spectra were measured and underwent deconvolution again with minimizing step size and increasing scanning number for a survey and Fe 2p, P 2p, and N 1s spectra. It was effective in reducing noises in the case of P 2p spectra for Fe-N-Cs, however, in the case of Fe, lower step size and more scan number were not valid in reducing noise.

Page 8, Line 23 - Page 9, Line 5;

XPS analysis for Fe-N-Cs was conducted again with a lower step size and higher scan number for higher accuracy.

References

[R1] Lee, S. H.; Kim, J.; Chung, D. Y.; Yoo, J. M.; Lee, H. S.; Kim, M. J.; Mun, B. S.; Kwon, S. G.; Sung, Y. E.; Hyeon, T. Design Principle of Fe–N–C Electrocatalysts: How to Optimize Multimodal Porous Structures? J. Am. Chem. Soc. 2019, 141 (5), 2035–2045. https://doi.org/10.1021/jacs.8b11129 

Round 2

Reviewer 1 Report

The authors revised their manuscript well. The revised version can be accepted for publication.
The distributions presented in Fig. 4b are disturbing. (a) a different shape of the PSD (w <1 nm) for sample Fe-N-C_PA-0.066 compared to the others. (b) no value 0 for w = 0.4 nm !!!!

Author Response

Reviewer 1

Before we respond to your valuable comment, we would like to inform you of a few corrections.

  1. An explanation of the definitions of Va and dVp/dW in Figure 4 has been added.
  2. The x-axis range in Figure 4B has been modified to avoid confusion.
  3. The manuscript was edited focusing on the effect of PA rather than the improvement catalytic activity.

-----------------------------------------------------------------------------------------------------------------------------------

The authors revised their manuscript well. The revised version can be accepted for publication.
Response: Thank you for your decision. We could develop our manuscript through your thoughtful comments.

  1. The distributions presented in Fig. 4b are disturbing. (a) a different shape of the PSD (w <1 nm) for sample Fe-N-C_PA-0.066 compared to the others.

Response: Thank you for your careful comment. In the case of PA-0.066, it was thought that the micropore distribution less than 1 nm were different as the PA did not sufficiently cover the carbon surface, resulting in non-uniform activation. We have tried to supplement the explanations for better understanding.

Page 7, lines 14-19;

For the PA-0.066, PA amount was not sufficient to cover entire carbon surface, therefore, partial activation of carbon surface induced the irregular activation of carbon, It was considered that the micropores could not be formed and rather collapsed and PA covered carbon underwent activation with pore expansion. Since PA was not uniformly distributed, therefore, it was expected that added PA excessively existed in some regions, resulted in pore expansion.

  1. (b) no value 0 for w = 0.4 nm !!!!

Response: Thanks for your valuable advice. Fig. We revised the x-axis of 4B and applied it to the manuscript.

Reviewer 2 Report

I appreciate that the authors have done additional measurements and added comparative literature results to improve their manuscript. However, I am now more convinced that the effects of PA doping on Fe-N-C is negligible in the current work.

  1. First, with the highest PA doping, the activity increases by only 10-15 mV which is not very significant over the pure Fe-N-C. Another concern is that looking at Figure 7c, the current work is far away from other literature results. If a work is an outlier, there should be strong evidence to suggest why it is so.
  2. The amount of P in the matrix now is consistent with amount of PA used. However, all other numbers are also similar, like N and C atomic percentages are similar. At the cost of which element is P increasing? I also fail to see how small changes of 0.3 at.% can cause significant improvement in catalytic performance. The activity values improvement of 10 mV is consistent with this small change – but I do not see this as a big improvement. Perhaps the original catalyst Fe-N-C is itself quite good?
  3. As the authors mention, P simply lies in the matrix without interaction with Fe. So, either there is no changes to the electronic state of Fe at all or below detection capabilities. I understand that the authors are unable to do in-depth electronic state characterisation due to lack of instrumentation. However, one can not speculate that the changes in activity could be due to presumed changes in electronic states, particularly when the changes in atomic percentage or ORR activity is not too different.
  4. Increasing PA interferes with graphitic structure but the effect on BET surface area and pore volume is negligible. As there are no error bars on values, I assume the surface area and pore volume values are all similar, i.e. within the error. This shows that PA treatment does not alter the structure of Fe-N-C significantly.
  5. The conclusion says, “ The changes in amount and size of micropores where remarkable…”. I do not agree with this as the changes are modest and therefore had no effect on ORR.
  6. Figure 4. Please define terms such as Va DVp/dW, W etc. in figure caption. Should the y-axis in figure 4b have a unit?

As mentioned, the starting catalyst Fe-N-C is good by itself. The effect of PA doping in enhancing activity is negligible. I suggest the authors to then present this as a simple study on the effects of PA doping rather than improvements, however any speculations must be avoided.  

Author Response

Reviewer 2

Before we respond to your valuable comment, we would like to inform you of a few corrections.

  1. An explanation of the definitions of Va and dVp/dW in Figure 4 has been added.
  2. The x-axis range in Figure 4B has been modified to avoid confusion.
  3. The manuscript was edited focusing on the effect of PA rather than the improvement catalytic activity.

-------------------------------------------------------------------------------------------------------------------------------------

I appreciate that the authors have done additional measurements and added comparative literature results to improve their manuscript. However, I am now more convinced that the effects of PA doping on Fe-N-C is negligible in the current work.

  1. First, with the highest PA doping, the activity increases by only 10-15 mV which is not very significant over the pure Fe-N-C. Another concern is that looking at Figure 7c, the current work is far away from other literature results. If a work is an outlier, there should be strong evidence to suggest why it is so.

Response: Thank you for your valuable comment. Generally, Fe-N-C catalysts often have a low Fe content of less than 3 wt.%, because Fe-N-C catalysts usually underwent heat treatment at high temperature and acid leaching process. Therefore, loading a large amount of catalyst on the electrode for half-cell measurement is frequently used to report high activity. However, in the actual operation of a single cell, cathodes with highly loaded Fe-N-C catalysts have disadvantages in terms of water transport due to electrode thickness, therefore, efforts have been made to solve the problem [R1]. Therefore, we think that catalytic activity with lower amount of catalysts needs to be reported to relate the half-cell and single-cell performance and further electrochemical characterization. For our study, if the loading amount is increased, it is expected to have similar activity to catalysts from references [71,76] of the manuscript. However, we consider that 240 μg/cm2 is the adequate loading amount for half-cell measurement and applied to our work.

  1. The amount of P in the matrix now is consistent with amount of PA used. However, all other numbers are also similar, like N and C atomic percentages are similar. At the cost of which element is P increasing? I also fail to see how small changes of 0.3 at.% can cause significant improvement in catalytic performance. The activity values improvement of 10 mV is consistent with this small change – but I do not see this as a big improvement. Perhaps the original catalyst Fe-N-C is itself quite good?

Response: Thank you for your advice. In the case of Fe-N-C, as previously reported by other research teams, the growth of Fe particles was suppressed through silica and the synthesis method was optimized, therefore, Fe-N-C has high ORR activity itself [R2]. In the case of P content, 0.3 at.% is a very small amount. However, the Fe content was actually about 0.6 at% converted from 3 wt.% Fe detected from ICP-OES. Of course, it is expected that the higher P content, the more effective the electronic state of Fe can be changed which might increase the ORR activity. However, in our study, when more PA was added, large iron phosphate particles were formed and the activity rather decreased. We do not want to insist that the method we report is the best method for doping P. However, we tried to focus on reporting the effect of PA on the pore structure or catalytic activity of Fe-N-C. We take your comments seriously and have modified the manuscript to explain the effects of PA rather than catalytic activity development.

  1. As the authors mention, P simply lies in the matrix without interaction with Fe. So, either there is no changes to the electronic state of Fe at all or below detection capabilities. I understand that the authors are unable to do in-depth electronic state characterisation due to lack of instrumentation. However, one can not speculate that the changes in activity could be due to presumed changes in electronic states, particularly when the changes in atomic percentage or ORR activity is not too different.

Response: We appreciate your insightful comment. We also revised the manuscript, considering that the context of our manuscript should be adjusted to the effect of PA rather than improvement of catalyst activity. And the speculation on electronic state of Fe was deleted.

  1. Increasing PA interferes with graphitic structure but the effect on BET surface area and pore volume is negligible. As there are no error bars on values, I assume the surface area and pore volume values are all similar, i.e. within the error. This shows that PA treatment does not alter the structure of Fe-N-C significantly.

Response: We appreciate your valuable advice. As you mentioned, there was no significant difference in the BET surface area. However, there are differences in pore volume and pore size distribution. Also, at Figure 3, it can be seen that the peak around 1° disappears or shifts due to the change of ordered pore structure of carbon in the low-angle part of XRD pattern. We thought that the above two show the change of the catalyst structure.

  1. The conclusion says, “The changes in amount and size of micropores where remarkable…”. I do not agree with this as the changes are modest and therefore had no effect on ORR.

Response: Thank you for your advice. We corrected the conclusion and wrote about the effects of PA.

  1. Figure 4. Please define terms such as Va DVp/dW, W etc. in figure caption. Should the y-axis in figure 4b have a unit?

Response: We are grateful for your suggestion. We added the definition of the term in Figure 4a. In the case of the y-axis of Figure 4b, it was left as it was thought to be necessary to quantify the change in pore volume by pore size.

As mentioned, the starting catalyst Fe-N-C is good by itself. The effect of PA doping in enhancing activity is negligible. I suggest the authors to then present this as a simple study on the effects of PA doping rather than improvements, however any speculations must be avoided.

Response: Thanks for your thoughtful comment. We revised the manuscript in the aspect of the effect of PA, agreeing that it is hard to emphasize that the effect of PA on improvement of catalytic activity.

References

[R1] Mustain, W.E.; Chatenet, M.; Page, M.; Kim, Y.S. Durability Challenges of Anion Exchange Membrane Fuel Cells. Energy Environ. Sci. 2020, 13, 2805-2838. 10.1039/D0EE01133A

[R2] Sa, Y. J.; Seo, D. J.; Woo, J.; Lim, J. T.; Cheon, J. Y.; Yang, S. Y.; Lee, J. M.; Kang, D.; Shin, T. J.; Shin, H. S.; Jeong, H. Y.; Kim, C. S.; Kim, M. G.; Kim, T. Y.; Joo, S. H. A General Approach to Preferential Formation of Active Fe-Nx Sites in Fe-N/C Electrocatalysts for Efficient Oxygen Reduction Reaction. J. Am. Chem. Soc. 2016, 138 (45), 15046–15056. https://doi.org/10.1021/jacs.6b09470 

Reviewer 3 Report

the manuscript may be accepted

Author Response

Reviewer 3

Before we respond to your valuable comment, we would like to inform you of a few corrections.

  1. An explanation of the definitions of Va and dVp/dW in Figure 4 has been added.
  2. The x-axis range in Figure 4B has been modified to avoid confusion.
  3. The manuscript was edited focusing on the effect of PA rather than the improvement catalytic activity.

------------------------------------------------------------------------------------------------------------------------

the manuscript may be accepted

Response: Thank you for your decision. We could improve our manuscript by your valuable comments.

Round 3

Reviewer 2 Report

The authors have explained the reasoning behind their methods well and I am happy for the paper to be accepted - changing my previous opinion. I also like the new conclusion; it reads better than before and reflects their study well. Best wishes.